# Driving Sustainable Innovation in New Ventures: A Study Based on the fsQCA Approach

Yu Liu [1] and Hao Zhang [2,*]

1  School of Business and Management, Jilin University, Changchun 130012, China; liuyu901129@163.com
2  Greater Bay Area International Institute for Innovation, Shenzhen University, Shenzhen 518061, China
*  Correspondence: zhanghaord@163.com; Tel.: +86-1580-430-4210

**Abstract:** With the external environment becoming increasingly complex and changeable, how we can effectively enhance the innovation of companies in sustainability has become the focus of research. For startups, due to their lack of resources and poor independent innovation capabilities, they need to search for external knowledge from outside to meet their own needs. Therefore, obtaining external knowledge sources and adopting appropriate methods for knowledge search is the key to affecting innovation in sustainability. Moreover, enterprise capability is also an important factor restricting sustainable innovation. In this paper, we construct an integrated framework of resources and capabilities based on theoretical learning and practice between 2018 and 2021, containing technical knowledge, market knowledge, a formal search, an informal search, organizational learning, and strategic flexibility. Taking 450 new ventures in China as the research sample, we adopt the fsQCA method and derive the path driving the sustainable innovation of new ventures. The results show that resources and enterprise capabilities combine to influence sustainable innovation, and there are two configuration paths driving the sustainable innovation of new ventures. In the first pathway, a technical knowledge search, a market knowledge search, organizational learning and strategic flexibility are the core conditions; in the second pathway, a formal search, an informal search, organizational learning, and strategic flexibility are the core conditions.

**Keywords:** sustainable innovation; knowledge search; organizational learning; strategic flexibility; new ventures; QCA





## 1. Introduction

With the COVID-19 pandemic sweeping across the globe, it has become increasingly difficult for companies to maintain sustainable innovation, and startups are the first to bear the brunt. Enterprise innovation in sustainability refers to continuously obtaining technical and market information, launching innovative products or service projects, and constantly obtaining economic, environmental, and social benefits [1]. By exploring innovation in sustainability, they can create dual value of economic and social sustainability [2]. Moreover, sustainable innovation also provides a new source for the sustainable development of enterprises [3]. Therefore, it is of great significance to explore what the important factors are in driving sustainable innovation and how to maintain sustainable innovation in new ventures. Given the lack of research on the integration of resources and capabilities [4–6], this paper attempts to select appropriate variables in terms of resources and capabilities, trying to explore how they work together to influence sustainable innovation. With the increasing complexity of technology, rising R&D costs, accelerating knowledge growth and diffusion, and shortening product life cycles, the external knowledge search has become the most important factor affecting sustainable innovation of enterprises [7–9]. Dynamic capabilities theory tells us that enterprises can rationally integrate and allocate resources through "enterprise capabilities" and should be able to cope with the external uncertain environments [4,10]. Organizational learning is the course of absorbing and processing new

knowledge [11], and it involves the recognition, screening, integration, and transformation of knowledge [12]. As one of the most important capabilities of an enterprise, it can provide a core competitive advantage [13]. However, for startups, due to their short establishment time and insufficient experience, they have not yet formed mature learning practices, and so it is not appropriate to apply traditional organizational learning theories. Therefore, it is necessary to explore the inner learning mode of new ventures and its effects on their sustainable innovation. Strategic flexibility is the special organizational ability of enterprises to react to changes in the external environment by reallocating organizational resources, processes, and management activities [14,15]. Enterprises with higher strategic flexibility can reduce their reaction time to environmental changes, and can redeploy resources more effectively, thereby increasing the value of innovative resources [15,16]. Therefore, strategic flexibility can effectively enhance the sustainable innovation of enterprises.

Following the study of Sofka and Grimpe [17] and Guo B and Guo JJ [18] concerning what to search for and how to search, this paper adds two capability variables and constructs an integrated framework of resources and capabilities. DZ Auto Equipment Manufacturing Company is a high-tech intelligent equipment manufacturing enterprise specializing in a robotic automatic welding production line, electrical automation, and robot application. During the survey, it was found that conducting external knowledge search activities and having better organizational learning and strategic flexibility are the keys to sustainable innovation. First, they obtained automation-related technologies and parts processing production lines from suppliers through search methods such as technology purchases and technology licenses, and obtained the core technologies of enterprise production; secondly, they cooperated with Jilin University, Chinese Academy of Sciences and other units to continuously learn new knowledge and upgrade core technologies to improve their own competitive advantages; finally, they were able to flexibly integrate and allocate resources within the enterprise, and respond quickly to changes in the external environment, which is also the key to maintaining competitiveness during the COVID-19 pandemic.

Indeed, sustainable innovation is a necessary condition for the application of new knowledge and can improve organizational learning capabilities while promoting economic growth of enterprises [19,20]. Therefore, sustainable innovation is the result of the combined effect of enterprise resources and capabilities.

Through a literature review, it is found that the following two aspects are mainly studied with regard to knowledge search. The first is the impact of knowledge search on innovation [21–23], while the second is the mechanism of knowledge search on innovation from different perspectives, such as the knowledge integration perspective [24,25], the absorptive capacity perspective [26–28], etc. Most studies used the dimensions of search width and depth, which cannot solve the problem of what knowledge enterprises search for and how to search for knowledge. To address this problem, we take a two-dimensional partitioning approach in our study for knowledge search. The search content (knowledge source) is divided into a technical knowledge search and market knowledge search, and the search method is divided into a formal search and an informal search.

In addition, most of the existing literature used regression empirical research methods to explore the relationship between resources and capabilities and enterprise sustainable innovation. However, this study method can only solve the marginal effect between variables and cannot solve the complex multivariate interaction [29]. Therefore, we take configuration as the study perspective, adopting the fuzzy set qualitative comparative analysis approach to incorporate the content attributes and method attributes of an external knowledge search, organizational learning, and strategic flexibility into a research model. In this way, it not only examines the impact of a single variable on sustainable innovation, but also explores the interaction between variables, and derives the way to drive sustainable innovation in new ventures.

Based on the analysis of the above arguments, we construct an integrated model of resources and capability, taking 450 new ventures in China as a research sample, and

conduct empirical analysis through the fsQCA approach. The results show that there are two configuration paths driving the sustainable innovation of new ventures. In the first pathway, a technical knowledge search, market knowledge search, organizational learning, and strategic flexibility are the core conditions; in the second pathway, a formal search, an informal search, organizational learning, and strategic flexibility are the core conditions.

Therefore, this paper makes the following related theoretical contributions. Our study's first contribution is constructing an integrated framework of resources and capabilities and exploring whether they are sufficient and necessary to drive sustainable innovation.

The second contribution is that it employs emerging management research methods and presents the configuration paths that drive the sustainable innovation of startups.

The third contribution is that it divides knowledge search into technical knowledge and market knowledge according to the search content, and divides knowledge search into formal search and informal search according to the search method. It also enriches the open innovation theory and expands the research boundary of knowledge search.

The aim here, therefore, is to answer three research questions: (1) Are knowledge search, organizational learning, and strategic flexibility the main factors and necessary conditions driving sustainable innovation? (2) Under the interaction of multiple variables, do resources and capabilities jointly drive the sustainable innovation? (3) What is the configuration path to enhance sustainable innovation in new ventures?

Based on the resources–capabilities integration view, this paper explores the factors that affect innovation for sustainable business and the configuration path that drives sustainable innovation in new ventures. The following section presents relevant theories and literature reviews on knowledge search, organizational learning, and strategic flexibility, which provides theoretical and literature support for our research on sustainable innovation. Next, we propose a conceptual model that drives sustainable innovation and tests the model with the fsQCA method. The final section provides conclusions from the study, managerial implications, and suggestions for sustainable innovation research in the future.

## 2. Theoretical Retrospection

The open innovation theory holds that the innovation of enterprises does not only rely on internal research and development, but more and more on external knowledge. Enterprises achieve innovation through the purposeful inflow and outflow of knowledge, thereby improving their sustainable competitive advantage in the market [30]. Therefore, enterprises can acquire the new knowledge they need for themselves through external knowledge search activities [21,31].

Nevertheless, the knowledge-based view believes that knowledge is an important resource for enterprises and can provide sustainable competitive advantages [32]. Knowledge management is characterized by the implementation of knowledge strategies and processes in an organization to increase the effectiveness and efficiency [33].

In addition, the dynamic capability theory holds that, facing the complexity and uncertainty of the external environment, enterprises need to cultivate their own dynamic capabilities. Strategic flexibility is a dynamic ability to flexibly allocate resources and coordinate processes when an organization faces environmental threats. Enterprises with high strategic flexibility can use their existing resources more flexibly, thereby increasing the innovative value of resources [15,34].

### 2.1. Open Innovation Theory

One of the most important theoretical backgrounds of open innovation is cooperative research and development. Before the rise of cooperative R&D, most of the R&D models followed the Schumpeter style, which was called "closed innovation" by Chesbrough [30]. By the end of the 20th century, the closed innovation model was gradually replaced by open innovation, and the new innovation process became more open, decentralized, and democratic, highlighting the interactive characteristics of the innovation process. Based on the open innovation theory proposed by Chesbrough [30], modern companies can not

only integrate beneficial creative resources from within, but also carry out cooperative innovation by introducing, digesting, and absorbing external diversified resources. This makes external knowledge search activities increasingly common.

Dahlander and Gann [35] summarized four reasons why open innovation is generally accepted:

(1) It reflects socioeconomic shifts in work patterns, with skilled workers seeking a portfolio of jobs rather than a lifetime of work;
(2) Globalization expands the scope of the market, thereby allowing a finer division of labor;
(3) An enhanced market system allows businesses to trade ideas;
(4) New technologies provide cooperation and coordination across geographic distances.

The existing research on open innovation is mainly carried out from two perspectives: the process perspective and the organizational perspective. Open innovation based on the process perspective mainly takes the input and output processes of innovation as a classification dimension. Von Hippel [36] classified innovation processes and outcomes into two dimensions; Knudsen and Mortensen [37] distinguished four types of open innovation with progressive characteristics. The classification of open innovation based on organizational perspective is richer and more diverse. Keupp and Gassmann [38] redefined open innovation by benchmarking high–medium–low portfolio differences in open innovation breadth and depth across firms.

Therefore, open innovation brings more external resources to enterprises, and also brings the possibility for enterprises to search for external knowledge.

### 2.2. Knowledge-Based View

The resource-based theory tells us that resources should be valuable, scarce, and irreplaceable [39,40]. Therefore, knowledge is the most important strategic resource of an enterprise. Knowledge can be defined as the assessment of an individual's learning resources and knowledge that allow them to record a good performance or task and indicate their sufficiency or suitability for it [41]. The enterprise is a knowledge processing system under the condition of bounded rationality and the specialized division of labor. The types of knowledge mainly include tacit knowledge and explicit knowledge [42–44]. Explicit knowledge is that which can be directly recorded, encoded, or easily communicated and transmitted. Tacit knowledge refers to knowledge that cannot be directly or easily transferred. Tacit knowledge can only be transferred slowly through "learning by doing", so it can bring a sustainable competitive advantage to a business [45,46].

Based on the knowledge-based theory, knowledge is the most important strategic resource of an enterprise, which can be created, stored, and applied, and can contribute to the sustainable innovation. This is mainly reflected in the following four aspects:

(1) A business is a knowledge system [47];
(2) Knowledge is the most important strategic resource of an enterprise [48];
(3) Companies can acquire and transform other resources by using knowledge [49];
(4) Knowledge possessed by firms is a key factor in making differences between them [50].

It can be seen that the ability of enterprises to use knowledge is the main reason for enterprises to maintain and enhance their competitive advantages. Therefore, only through learning can an enterprise transform the new knowledge acquired from the outside into the competitiveness.

### 2.3. Dynamic Capability Theory

Enterprise capability is an important strategy, which comes from learning, the acquisition of external resources, and the reallocation of resources in addition to individual capabilities [51]. Makadok [52] defined capability as a special type of resource whose purpose is to increase the productivity of other resources owned by a firm; thus emphasizing the distinction between capability and resource.

According to the dynamic capability view, the term "dynamic" refers to the ability to update existing capabilities to achieve flexibility in response to changing circumstances. The term "capability" emphasizes the need for strategic management to properly adjust, integrate, and reallocate internal and external organizational resources and capabilities to adapt to changing circumstances [4,53].

There are two main research perspectives on dynamic capabilities: the cognitive perspective and the process perspective. Scholars with a cognitive perspective argue that firms with highly dynamic capabilities can identify threats and opportunities, influence external changes that align an organization with its business environment, and prevent the emergence of organizational rigidity or inertia [10,51]. Scholars who hold a process perspective view dynamic capabilities as firm practices or processes, and they see dynamic capabilities as a tool that can exist in the form of specific and identifiable processes [4].

In addition, the current background of the digital economy, open innovation, "Internet+" and the COVID-19 pandemic have endowed enterprises with new situations of dynamic capabilities, and the study of dynamic capabilities in special situations has also become a hot spot at this stage.

### 2.4. External Knowledge Search

As technology becomes increasingly complex, R&D costs continue to rise, and product life cycles continue to shorten, traditional "closed innovation" can no longer meet the knowledge needs of these startups. As a result, they increasingly emphasize the importance of external knowledge, and generally use external knowledge sources. By combing through the existing literature, we have found that the related research of external knowledge search mainly focuses on the following aspects: the first is the emphasis of external sources of knowledge, such as technology purchases, R&D cooperation, alliances, etc. [54,55]. The second is the emphasis of influencing factors of knowledge search, such as the network environment, previous experience, innovation objectives, etc. The third is the study of relationship between the external knowledge search and performance. For example, Wang et al. [8] took data from 187 corporate entities as a research sample, and divided knowledge search into two dimensions of search width and search depth and built a relationship model between knowledge search and organizational innovation performance. Although the research on knowledge search is increasingly rich, more research needs to be done. One reason for this is that the research boundary of knowledge search is still very vague. Some scholars believe that knowledge search is a process of searching for and acquiring knowledge from external knowledge sources. However, some scholars argue that knowledge search includes not only the acquisition process, but also the knowledge integration and creation. The second is that, in existing research, the division of knowledge search dimensions mainly adopts the research results of Laursen and Salter [21], who divided knowledge search into search width and search depth. However, the knowledge search is a complicated process. Questions such as what knowledge to search for and how to search for it need to be solved urgently. The third is that the past research objects were mainly mature enterprises. These authors believed that the relationship between knowledge search and innovation was an "inverted U-shaped" one and concluded that the search was excessive. However, due to a lack of resources, startups have a more obvious dependence on external resources, and so it is more meaningful to take startups as research objects. Based on this, our research constructs a multi-dimensional knowledge search framework, divided into technical knowledge search and market knowledge search in accordance with the search content, and divided into a formal and informal search in accordance with the search method, as is shown in Figure 1.

In future research, first, it is necessary to further explore the research boundary of external knowledge search, and build a new research perspective in the context of the new era, such as the digital economy and the COVID-19 pandemic; secondly, it is necessary to further explore issues related to external knowledge search for small and medium-sized enterprises and start-ups, as well as their differences and connections with

mature enterprises; thirdly, it is necessary to examine the influence and mechanism of the combination and balance of different dimensions of knowledge search on innovation.

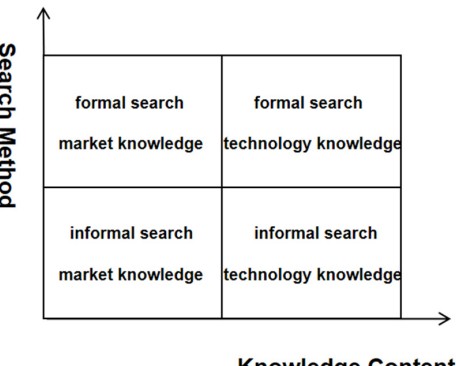

**Figure 1.** Knowledge search framework. Note: Figure 1 is modified and proposed by the author according to the views of Sofka and Grimpe [17] and Guo B and Guo JJ [18].

### 2.5. Organizational Learning

The theory of organizational learning was first put forth by Argyris and Schon, who considered it as the result of members interacting within the internal environment of the enterprise [56,57]. In accordance with earlier studies, scholars believed that organizational changes driven by management issues are the key to learning [58]. By the mid-1990s, people generally believed that various organizations could adapt to the changing environment through learning, learn from past successes or failures, and foresee and respond to upcoming threats [59,60]. Through the research on organizational learning, it was found that organizational learning generally has two research perspectives: the ability view and the process view. Scholars who hold the concept of competence believe that organizations need to respond quickly to changes in the external environment and strive to become a learning organization [61]. Scholars who hold a process view believe that organizational learning is a process of knowledge processing by an organization, including knowledge acquisition, knowledge integration, knowledge allocation and utilization, and knowledge creation [62].

However, the existing research still has some limitations. For example, in the early studies, only knowledge management processes were analyzed in intensive departments, and the acquisition, sharing, utilization, creation, and storage of knowledge were not studied as a whole framework. In addition, although the organizational learning mode and framework for large-scale enterprises are relatively mature, organizational contexts, learning approaches, and entrepreneurial roles in startups are different; therefore, the original mature theory cannot be fully applied. As a result, this article explores the internal mechanism of organizational learning by taking startups as the research object.

For startups, organizational learning is a multi-level dynamic process that integrates psychological and social processes [63]. It includes knowledge absorption and integration by members in the organization, the transfer and sharing of knowledge, the application of knowledge, the creation of knowledge, and the storage of new knowledge, as shown in Figure 2. Knowledge recognition involves screening the inflow of new knowledge and integrating it with the internal understanding of the enterprise. In this way, the enterprise can quickly identify the nature of knowledge and shorten the knowledge distance. The essence of knowledge sharing in startups is the flow of knowledge among employees and between employees and entrepreneurs. Through knowledge sharing, members of different departments can exchange knowledge, which can improve the overall knowledge level. Knowledge integration is the process of reallocating the internal knowledge of the enterprise. The integrated knowledge will have the characteristics of the actual new understanding and some differences, and this kind of knowledge bears the unique context of the organization. After identifying, sharing, and integrating knowledge, enterprises can

fully absorb the acquired new knowledge and create new knowledge. Then, the enterprises need to perform the explicit processing and storage of new knowledge to increase the firm's total amount of knowledge, and the spillover effect of knowledge can also bring new opportunities for the enterprises.

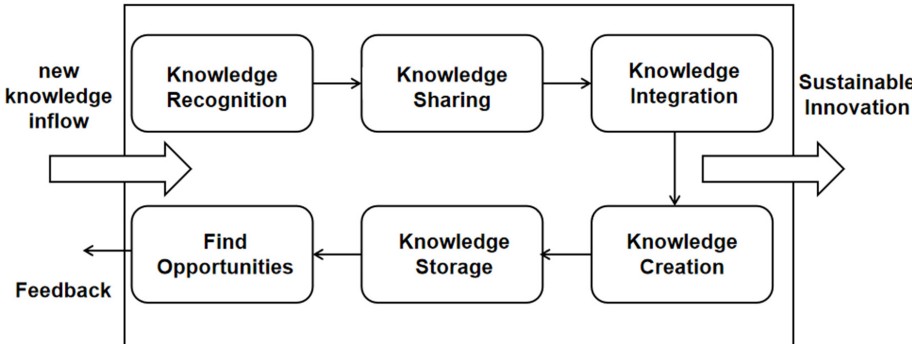

**Figure 2.** Internal framework of organizational learning. Note: Figure 2 is the author's point of view based on the above literature and related theories.

The first aim is to continue to explore the theoretical framework and process of organizational learning, and the main research objects are small and medium-sized enterprises and new ventures. As small and medium-sized enterprises and new ventures are more dependent on external knowledge, it is crucial for them to organize learning internally, especially with new and emerging ways of learning. The second is the division of organizational learning dimensions from various perspectives and their impact on innovation. Due to changes in the environment, learning strategies will be continuously adjusted, and the dimensions of organizational learning will be more diverse. The existing empirical research conclusions are inconsistent, and so it is necessary to continue to explore the relationship between organizational learning and innovation, especially with SMEs and new ventures as the research objects.

### 2.6. Strategic Flexibility

Strategic flexibility originated from the idea of "mutation management" and gradually coordinated with the resource-based view and contingency theory. In the 1980s, strategic flexibility helped in the first stage of development, emphasizing corporate strategic choices and options. At this stage, the traditional business assumptions were surpassed, and the concept and model of strategic learning were constructed. The second stage of the development of strategic flexibility was after the 1990s, reconsidered from the perspective of resources and capabilities. Compared with the previous stage, this stage originated from the fuzzy cognition of strategic flexibility. From the perspective of resources and capabilities, it explains why strategic flexibility can help enterprises deal with the threats of the external environment.

Strategic flexibility is a special organizational ability that an enterprise achieves to respond to changes in the external environment by re-allocating organizational resources, processes, and management activities [15,64]. Strategic flexibility can be proactive, for example, by sensing and adapting to changing customer needs in the market or by reacting to the emergence of new competitors [15]. Therefore, enterprises with strategic flexibility can reduce the response time to dynamic changes and can redeploy resources more effectively, thereby increasing the value of innovative resources [15,34].

However, although most studies have shown that strategic flexibility can positively impact the enterprise's competitiveness and innovation capabilities [65], the consensus on the relationship between strategic flexibility and corporate competitiveness has not been fully agreed upon, especially among small and medium-sized enterprises. The research on strategic flexibility still faces challenges and obstacles. Some scholars believe that the size of an enterprise affects the relationship between strategic flexibility and sustainable

innovation [66]. Compared with large enterprises, startups face many difficulties in fiercely competitive fields, and they are more likely to be exposed to unfamiliar environments. Some scholars believe that the impact of strategic flexibility on sustainable innovation was not significant [67], and some even came to the opposite view [68]. Therefore, the balance between the pros and cons of strategic flexibility in startups needs to be explored urgently.

## 3. Model

A technical knowledge search refers to the behavior of enterprises by crossing their organizational boundaries to search for and obtain knowledge related to product design, technology, and processes from outside [69,70]. Technical knowledge mainly comes from universities, scientific research institutions, suppliers, and other external enterprises. First of all, an enterprise can obtain the relevant technology needed by the technology purchasing, technology licensing, cooperative research and development, etc., which can be used to correct the technical defects of its products and improve the technical content of the enterprise. The new technical knowledge acquired by the enterprise from outside can be matched and integrated with the original knowledge to produce new knowledge to improve the innovation. Secondly, the scattered and fragmented technical knowledge acquired by the enterprise from the external network can be transformed into new knowledge to improve the level of technology [32,71]. In addition, through technical knowledge search activities, enterprises can greatly increase the knowledge reserves of their internal knowledge base, and at the same time improve their ability to allocate technical resources, which can help to better enhance their sustainable innovation.

A market knowledge search refers to the behavior of enterprises in searching for and obtaining knowledge related to product design solutions, marketing channels and business models by crossing the organizational boundaries, etc. [70]. Market knowledge mainly comes from competitors and customers in the same industry [21]. Market knowledge can also provide benefits for the sustainable innovation of startups. First of all, by establishing contact with customers, it is possible to get feedback on the performance, quality, sales status, and other factors of the enterprise's products, which is conducive to the improvement in knowledge and the increase in its market share. Secondly, the market knowledge acquired from competitors can enable enterprises to understand their own advantages and disadvantages, so as to develop choices and strategies for future knowledge acquisition. Finally, obtaining a large amount of market information through market research, information feedback, etc., can enable enterprises to fully realize market the situation of market complementary products and substitutes, and promote knowledge overflow.

A formal knowledge search refers to the behavior of establishing formal contact with external organizations and acquiring knowledge through contracts, agreements, etc., such as technology purchasing, technology licensing, R&D cooperation, technology alliances, etc. [18]. A formal knowledge search will affect the enterprise's sustainable innovation according to the following aspects: first, through a formal knowledge search, the enterprise's explicit knowledge will be increased, which will help improve the enterprise's knowledge stock and absorptive capacity. Second, establishing an excellent formal relationship with external organizations can increase mutual trust between organizations, establish a mutual learning mechanism, and reduce costs in subsequent searches [72].

Informal knowledge search refers to the behavior of acquiring knowledge through informal communication, exchanges, etc., such as private meetings, conversations, informal employment, reverse engineering, etc. [70]. Informal knowledge search will affect the sustainable innovation of enterprises according to the following aspects: First, employees of an enterprise can establish "weak relationships" by interacting with customers, suppliers, or competitors through private communication and other methods. By establishing this "weak relationship", the breadth of the knowledge search can be increased, thereby obtaining more required knowledge and saving costs. Second, through an informal search, it is more conducive for employees to understand tacit knowledge. Since tacit knowledge is less likely

to be imitated by other competitors, it makes enterprises more conducive to maintaining a sustainable competitive advantage.

It is obvious that knowledge resources constitute the most important part of organizational learning [73], which combines personal judgment, values, abilities, know-how, and technology [74]. Knowledge management is characterized by the implementation of knowledge strategies and processes in the organization to improve the effectiveness and efficiency of business processes and to maintain organizational innovation [60]. Therefore, an organization can provide itself with a core competitive advantage through various knowledge activities. First, personal learning can improve professional skills and practical experience. Entrepreneurs are the core leadership of startups. They can constantly imitate and reflect through learning methods such as experiential learning and cognitive learning, which can reduce the risk of the enterprise in its growth process, they can promote the formation of the enterprise's future innovation strategy, and they can continue to carrying out opportunity identification and utilization. Second, organizational learning can promote the transfer and sharing of knowledge. Organizational decision making, leadership, problem-solving speed, and innovation ability are all improved by spreading knowledge across the parties [75]. Finally, new knowledge can be created through organizational learning. Knowledge creation is characterized by cultivating new abilities and expertise within the organization [76]. Startups can effectively maintain their competitive advantage by producing new knowledge and improving innovation capabilities.

Strategic flexibility reflects the ability of enterprises to flexibly use and allocate resources in response to environmental changes [15]. In today's rapidly changing environment, enterprises need to be able to quickly invest resources and use them freely when responding to changes—that is, to have strategic flexibility [77]. Strategic flexibility allows firms to easily devote resources to different products, regroup resources for production, redefine products and markets, reallocate resources to support new product strategies, etc. [15]. Therefore, the improvement in strategic flexibility can provide sustainable competitive advantages and innovation [78,79]. This is mainly reflected in the following two aspects: First, through the flexible allocation and use of resources, strategic flexibility can meet the needs of innovation for resources, so that enterprises can change key elements or the structure connecting these elements, and then carry out innovation activities [80,81]. Second, strategic flexibility enables enterprises to flexibly allocate and utilize resources and can motivate enterprises to innovate in order to realize the value of internal resource endowments [15].

In summary, this research adopts the open innovation theory and the dynamic capability theory. It takes configuration as the research perspective to construct a framework for the effects of external knowledge search, organizational learning, and strategic flexibility on sustainable innovation in new ventures. The specific model is shown in Figure 3.

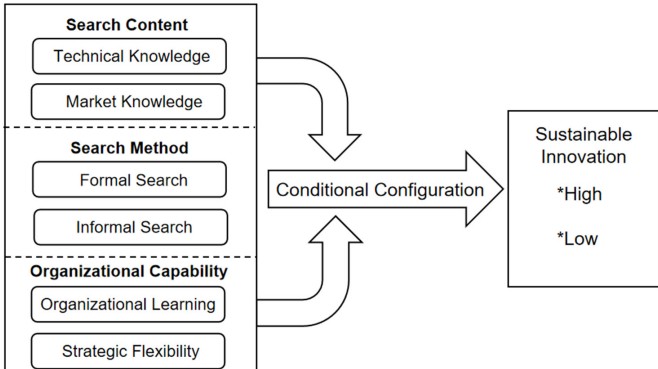

**Figure 3.** Causal configuration analysis model. Note: The author proposes this model based on the above literature and theory; "*High" represents the promotion of Sustainable Innovation; "*Low" represents a reduction of Sustainable Innovation.

## 4. Data and Methods

### 4.1. Research Methods

Charles Ragin first proposed qualitative comparative analysis (QCA). It is a research method between qualitative case analysis and quantitative statistical analysis and has the advantages of both [82]. Its main idea is to apply set theory and Boolean algebra to discuss the influence of conditional variables on result variables under configuration matching. Since this method is not limited by sample size, it was first applied to the field of sociology and political science [83]. When Fiss published an article on the feasibility of QCA in organizational strategy in the "Academy of Management Review" in 2007, the QCA method was brought into the field of management. Now, with the continuous optimization of this method, it has become an important tool for research in various management fields [84]. FsQCA is a more recent and extended version of the QCA that may be used for contextual analysis, which investigates how causal relationships are dependent on contextual conditions and is furthermore much closer to statistical approaches [85,86]. This article chose the fsQCA method mainly for the following reasons: One is that the traditional statistical analysis method solves the causal relationship between variables, ignoring the mutual influence between the independent variables. The second is that QCA advocates causal asymmetry and can simultaneously propose two paths of high-level configuration and low-level configuration. Third, compared to the two research methods of CsQCA and mvQCA, the fuzzy set has both qualitative and quantitative attributes, and it can distinguish them accurately [87].

### 4.2. Samples and Data

(1) Sample selection and collection. Based on the research purposes, this article focuses on startups in China. First of all, we conducted a two month follow-up survey at the DZ Automotive Equipment Manufacturing Company in the summer of 2018 (once a week on average) and conducted interviews and exchanges with the company's general manager and other department heads to understand the company's knowledge needs, knowledge search methods, market operation models, and a series of problems at this stage, laying a practical foundation for this research. Then, from September 2019 to March 2020, we performed an in-depth examination of the KD technology business incubator to conduct research and interviews with the executives. We came to understand the current situation of incubated enterprises, and understand their knowledge sources and methods, as well as the learning process, which provides important practical support for this study. Finally, our questionnaire was distributed and collected from September 2020 to March 2021. According to the relevant research of "Global Entrepreneurship Monitor- China Report", it was divided into areas with developed entrepreneurial activities and areas with less developed entrepreneurial activities. Therefore, this study selected enterprises in Shenzhen, Guangzhou, and Beijing as representatives in the entrepreneurship-developed areas, and selected enterprises in Changchun and Shenyang as representatives in the less-developed entrepreneurship areas, which is basically applicable to new ventures in China. Offline, we collected questionnaires through technology parks, incubated enterprises, talent markets, and university seminars, while online, we delivered questionnaires by email and contacted the local alumni association for them to fill in. The reasons for choosing new ventures as the research object were as follows. First, the internal resources of new ventures are more scarce, and they are more dependent on external resources, and the knowledge search activities are more obvious; second, as an emerging market in China, startups occupy a higher proportion; and third, with the intensification of the COVID-19 pandemic, they are facing greater challenges, and so it is more meaningful to study new ventures.

A total of 1036 questionnaires were issued this time, and 545 questionnaires were collected. After excluding those invalid questionnaires with missing data or inadequate answers, a total of 450 valid questionnaires were obtained, accounting for 43.4% of the total number of questionnaires issued. For details, see Table 1.

**Table 1.** Basic features of the questionnaire.

| Feature | Items | Quantity | Percentage |
|---|---|---|---|
| Regional source | Entrepreneurial developed areas | 339 | 75.3% |
| | Entrepreneurial underdeveloped areas | 111 | 24.7% |
| Questionnaire Features | Questionnaires issued | 1036 | 52.6% |
| | Questionnaire recovery | 545 | |
| | Questionnaire valid | 450 | 43.4% |

(2) The statistical characteristics of the sample. The number of valid samples from Shenzhen, Guangzhou, and Beijing was 339, accounting for 75.3% of the effective sample number, and the number of valid samples from Changchun and Shenyang was 111, accounting for 24.7%. In terms of corporate age, 211 enterprises were less than 5 years old, accounting for 46.9% of the total, and 239 enterprises were 6–8 years old, accounting for 53.1% of the total. In terms of enterprise scale, there were 344 firms with less than 500 employees, accounting for 76.5% of the total, and 106 enterprises with more than 500 employees, accounting for 23.5%. In terms of industry, there were 148 enterprises in the manufacturing industry, accounting for 32.9% of the sample; there were 151 enterprises in the information/computer/software industry, accounting for 33.6% of the total sample; there were 41 enterprises in the retail industry, accounting for 9.1% of the total sample; there were 70 enterprises in the service industry, accounting for 15.6% of the total sample, and there were 40 enterprises in other industries, accounting for 8.9% of the total sample.

Specific sample characteristics are shown in Table 2.

**Table 2.** Sample characteristics.

| Characteristics | Percentage | Characteristics | Percentage |
|---|---|---|---|
| Age | | Scale | |
| 1–2 years | 18.9% | 1–20 | 3.1% |
| 3–5 years | 28.0% | 21–50 | 10.2% |
| 6–8 years | 53.1% | 51–100 | 25.6% |
| Industry | | 101–200 | 16.0% |
| Manufacturing | 32.9% | 201–500 | 21.6% |
| Information industry | 33.6% | 501–1000 | 11.3% |
| Retail industry | 9.1% | More than 1000 | 12.2% |
| Service industry | 15.6% | | |
| Other | 8.9% | | |

*4.3. Variables*

The condition variables used in this study are external knowledge search, organizational learning, and strategic flexibility, and the result variable is the sustainable innovation. In the questionnaire design process, the following principles were mainly followed: one was to choose a scale that has been cited many times; the second was to adopt a scale with higher reliability and validity; and the third was to choose a widely used scale. First of all, relying on the focal discussion of the original scale of our study by the teachers and doctoral students of our team at the regular academic meeting, we revised the expression content of the items in the scale combined with the topics to be studied, and added and deleted some items to form the scale used in the study, and constructed a research questionnaire. Then, we communicated with entrepreneurs and executives of some new enterprises on the content of the scale and asked them to fill out the questionnaire to see whether each item is clearly expressed with regard to semantics and whether there are any comprehension issues. At the same time, we tested whether each dimension was significant. This research used the 7-point Likert scoring method, where "1" represents completely inconsistent with the item and "7" represents completely consistent with the item. The following is the measurement of each variable in this study.

(1)  Sustainable innovation (SI). This article mainly drew on the research of Story, Boso, and Cadogan [88] and Hansen and Birkinshaw [89], involving a total of 6 items

such as the innovation ability of the enterprise and the satisfaction of the product or service. Examples include "We have a strong ability to innovate", "Compared with competitors, our products or services have more advantages", and so on. The Cronbach's alpha of this variable was 0.803, and the AVE was 0.54, indicating that it reached acceptable standards in terms of reliability and validity.

(2) External knowledge search. In terms of a technical knowledge search (TKS) and market knowledge search (MKS), this research referred to the research of Sofka and Grimpe [17] and Ki H and Jina [55]. The formal search (FS) and informal search (IS) are based on the research of Guo B and Guo JJ [18]. Examples include "We are good at seeking technical knowledge through purchasing technology, licensing, etc.", "We are good at seeking market information such as new sales channels and marketing strategies from enterprises in the same industry through formal cooperation", "We are good at imitating competitors in the same industry and improving technology", and "We are good at seeking product or service-related market information from customers through informal communication". The Cronbach's alpha of each variable was 0.830, 0.827, 0.816, and 0.803, and the average was 0.40, 0.49, 0.45, and 0.54, indicating that the reliability and validity reached acceptable standards.

(3) Organizational learning (OL). This article drew on the research of Morales et al. [90], involving five items including knowledge acquisition ability and knowledge sharing. Examples include "The new knowledge or skills we acquire can bring a competitive advantage to the company" and "We are good at sharing and communicating knowledge to improve our skills." The Cronbach's alpha of this variable was 0.772, and the AVE was 0.53, indicating that it reached acceptable standards in terms of reliability and validity.

(4) Strategic flexibility (SF). This paper drew on the research of Zhou and Wu [15] and divided strategic flexibility into two dimensions—resource flexibility and coordination flexibility. Examples include "We flexibly allocate production resources to manufacture various products" and "We will effectively redeploy organizational resources to support the firm". The Cronbach's alpha of this variable was 0.791 and the AVE was 0.54, indicating that it reached acceptable standards in terms of reliability and validity.

## 5. Results

### 5.1. Data Calibration

It is necessary to calibrate the sample data of the result variable and the condition variable before the QCA test. The so-called data calibration involves transforming the existing raw data into fuzzy-set data required for QCA analysis. In addition to constructing total membership points and non-membership points in this study, it is also necessary to build crossover points [83]. Because the mean value represents the degree of aggregation of the data, and the standard deviation represents the degree of dispersion of the data, according to the research method of Kraus et al. [91], the full membership point is set as "the sum of the mean value and a standard deviation", the non-membership point is set as "the difference between the mean and one standard deviation", and the crossover point is set to the mean value. The calibration results are shown in Table 3. It can be seen from the results that the consistency level of all the condition variables is less than 0.9, and so there is no necessary condition.

**Table 3.** Variable calibration results.

| Variable | Full Membership | Crossover Point | Non-Membership |
|---|---|---|---|
| SI | 6.1296 | 5.2937 | 4.4578 |
| TKS | 6.0416 | 5.3873 | 4.7330 |
| MKS | 6.1008 | 5.4087 | 4.7166 |
| FS | 6.0969 | 5.4894 | 4.8819 |
| IS | 6.0752 | 5.3030 | 4.5308 |
| OL | 6.2748 | 5.4818 | 4.6888 |
| SF | 6.1417 | 5.3515 | 4.5614 |

## 5.2. Necessity Analysis

Necessity analysis mainly examines the extent to which it constitutes a subset of the condition. If a particular condition variable always appears in the configuration path, it is of necessity, and the measure of the necessity is consistency, the minimum standard of which constitutes a necessary condition, generally considered to be 0.9 [92]. Table 4 shows the conditional necessity test results of conditional variables and outcome variables.

**Table 4.** Conditional necessity test results.

| Variable | High-SI | | Low-SI | |
|---|---|---|---|---|
| | **Consistency** | **Coverage** | **Consistency** | **Coverage** |
| TKS | 0.7744 | 0.7847 | 0.4509 | 0.4093 |
| ~TKS | 0.4169 | 0.4588 | 0.7627 | 0.7518 |
| MKS | 0.7635 | 0.7817 | 0.4546 | 0.4170 |
| ~MKS | 0.4307 | 0.4686 | 0.7620 | 0.7426 |
| FS | 0.7699 | 0.7818 | 0.4586 | 0.4172 |
| ~FS | 0.4260 | 0.4677 | 0.7601 | 0.7579 |
| IS | 0.7824 | 0.7849 | 0.4613 | 0.4145 |
| ~IS | 0.4164 | 0.4631 | 0.7605 | 0.7579 |
| OL | 0.7898 | 0.7741 | 0.4888 | 0.4291 |
| ~OL | 0.4175 | 0.4769 | 0.7426 | 0.7599 |
| SF | 0.8057 | 0.7877 | 0.4729 | 0.4141 |
| ~SF | 0.4007 | 0.4591 | 0.7577 | 0.7775 |

Notes: "~" represents the logical operation "not". "High-SI" represents the path to enhance sustainable innovation; "Low-SI" represents the path to reduce sustainable innovation.

## 5.3. Causal Configuration Analysis

Configuration analysis attempts to expose the causal relationship of multiple conditional variables. In constructing the truth table, the adequacy analysis of the configuration is required. According to the research ideas of Ragin [82], this paper sets the case selection frequency to "1", with the consistency level being greater than 0.8, and the PRI consistency level being greater than 0.75. In the end, we obtain parsimonious, intermediate, and complex solutions. In this paper, the intermediate solution is used as the main reference, and the parsimonious solution is used as an auxiliary reference. As a consistent conclusion of the relationship between the conditional variables and the outcome variables in this paper has not been reached in the academic world; in the analysis process, this article will include all six variables in the "may exist/not exist" option, and analysis by using the software fsQCA3.0 yields the results of the configuration path analysis, as shown in Table 5.

Through the above configuration analysis results, it is found that the solution consistency of high sustainable innovation is 0.8978, which is greater than the acceptable standard of 0.85, and the solution coverage is about 0.64, indicating that a total of 64% of the sample companies are explained. Therefore, these two configuration paths can be used as the driving conditions for the sustainable innovation of the startups. The solution consistency of low sustainable innovation is 0.8768, which is greater than the acceptable standard of 0.85, and the solution coverage is about 0.52, indicating that a total of 52% of the sample enterprises are explained. Therefore, these two paths can be used as a combination of conditions that hinder the sustainable innovation of startups.

**Table 5.** Configuration path analysis results.

| Variable | High-SI | | Low-SI | |
|---|---|---|---|---|
| | **Path1** | **Path2** | **Path1** | **Path2** |
| TKS | ● | • | | ⊗ |
| MKS | ● | • | ⊗ | ⊗ |
| FS | • | ● | ⊗ | ⊗ |
| IS | | ● | ⊗ | |
| OL | ● | ● | ⊗ | ⊗ |
| SF | ● | ● | ⊗ | ⊗ |
| Consistency | 0.9167 | 0.9128 | 0.8767 | 0.8814 |
| Coverage | 0.5880 | 0.5786 | 0.5049 | 0.5043 |
| Solution consistency | 0.8978 | | 0.8768 | |
| Solution coverage | 0.6434 | | 0.5188 | |

Notes: ● = core causal condition (present), implying that the presence of the condition is crucial to the outcome; ⊗ = core causal condition (absent), suggesting that the absence of the condition is crucial to the outcome; • = contributing causal condition (present), implying that the presence of the condition is not essential to the outcome; ⊗ = contributing causal condition (absent), suggesting that the absence of the condition is not essential to the outcome; blank spaces indicate that the presence or absence of the condition does not matter with regard to the outcome. "High-SI" represents the path to enhance sustainable innovation; "Low-SI" represents the path to reduce sustainable innovation.

It can be seen from the analysis results that there are two configuration paths for driving high levels of sustainable innovation. In configuration path H1, the technical knowledge search, market knowledge search, organizational learning, and strategic flexibility are the core conditions, and the informal search is the edge condition. Therefore, this paper summarizes this path as a knowledge content-driven path. In configuration path H2, the formal search, informal search, organizational learning, and strategic flexibility are the core conditions, and the market knowledge search is the edge condition. This paper summarizes the path as a search method-driven path.

(1) Content-driven path (TKS × MKS × OL × SF). This configuration path emphasizes the importance of external knowledge sources. Moreover, the consistency between the condition variables and sustainable innovation is between 0.7 and 0.9, see Table 4. Therefore, we believe these variables are sufficient conditions to affect sustainable innovation. First, new ventures applying this driving path emphasize the importance of technical knowledge and market knowledge in terms of resources. This is consistent with studies by Voss [93] and Sofka and Grimpe [17]. They believed that technical knowledge is a key factor driving product development and innovation [17,46]. The emergence of new technologies will promote the upgrading of products, so that enterprises can gain greater competitive advantages and occupy a higher market share [94]. However, focusing only on technical knowledge and ignoring market knowledge is not advisable. After the enterprise has mastered the relevant market development trend and demanded change information, it can obtain various technical knowledge required for business development in time to protect the existing market position [95]. Thus, new ventures promote sustainable innovation with the combined effect of technical knowledge and market knowledge. Second, new ventures that apply this driving path emphasize the importance of organizational learning and strategic flexibility in terms of capabilities. This is consistent with the results of Liao et al. [96] and Sanchez [97]. They believe that the level of learning ability determines the effect of knowledge utilization, and the application of new knowledge can enhance sustainable innovation among enterprises [96]. In addition, the enterprise has high strategic flexibility, can better integrate and allocate the internal resources, and can cope with the changing external environment [14,97]. To sum up, this configuration path can promote sustainable innovation in new ventures through the resources–capabilities integration.

(2) Method-driven path (FS × IS × OL × SF). This configuration path emphasizes the importance of searching methods. Moreover, the consistency between the condition variables and sustainable innovation is between 0.7 and 0.9, see Table 4. Therefore, we believe these variables are sufficient conditions to affect sustainable innovation.

First, new ventures applying this driving path emphasize the importance of formal and informal searches in terms of resources. They focus more on knowledge search methods than knowledge sources. This is consistent with the research of Guo B and Guo JJ [18]. They argued that alliances, R&D cooperation, informal exchanges, hiring employees, reverse engineering, professional knowledge training, technology licensing, and patent purchases are all important ways for companies to acquire knowledge resources [18]. Through formal search activities, companies can quickly acquire the knowledge they need and build inter-organizational trust to foster sustainable innovation [98]. An informal search is more conducive to the transfer of tacit knowledge, and the cost is lower [69,99]. In addition, new ventures applying this driving path also emphasize the importance of organizational learning and strategic flexibility in terms of capabilities. In short, through the resources–capabilities integration, this configuration path can also promote sustainable innovation in new ventures.

In addition, two configuration paths have also been derived from this research that lead to reducing sustainable innovation in new ventures, as shown in Table 4. In the configuration path L1, ~FS, ~IS, ~OL, and ~SF are the core conditions, and ~MKS is the edge condition. It can be seen that the lack of knowledge search methods and the lack of organizational learning and strategic flexibility are the reasons for the low sustainable innovation levels of startups. In the configuration path L2, ~TKS, ~MKS, ~OL, and ~SF are the core conditions, and ~FS is the edge condition. It can be seen that insensitivity to the knowledge they need and the lack of organizational learning and strategic flexibility are further reasons for the low sustainable innovation of startups.

### 5.4. Robustness Test

This study conducted robustness testing by changing the consistency threshold and the number of samples [100,101] to ensure the robustness that drives high sustainable innovation. First, the consistency threshold was increased from 0.8 to 0.85, the sample frequency remained unchanged, and the result did not change compared to before the adjustment.

Second, this study used random sampling to select general samples for re-testing, and the results are shown in Table 6. Comparing Tables 5 and 6, we can see that there are still two paths to drive high levels of sustainable innovation, and the core conditions that constitute this path are roughly the same. Furthermore, their level of consistency and coverage is not very different than before. Therefore, this study can be considered to have good robustness.

**Table 6.** Robustness test results.

| Variable | High-SI | |
| :---: | :---: | :---: |
| | **Path1** | **Path2** |
| TKS | ● | ● |
| MKS | ● | |
| FKS | ● | ● |
| IKS | | ● |
| OL | ● | |
| SF | ● | ● |
| Consistency | 0.9266 | 0.9370 |
| Coverage | 0.6398 | 0.6171 |
| Solution consistency | 0.9268 | |
| Solution coverage | 0.6657 | |

Notes: ● = core causal condition (present), implying that the presence of the condition is crucial to the outcome; ● = contributing causal condition (present), implying that the presence of the condition is not essential to the outcome; blank spaces indicate that the presence or absence of the condition does not matter with regard to the outcome.

## 6. Conclusions, Implications, and Limitations

### 6.1. Conclusions

Based on the open innovation theory and dynamic capability theory, this paper took 450 startups in Shenzhen, Guangzhou, Beijing, Changchun, Shenyang, and other regions of China as the research sample. It took the configuration perspective and used the fsQCA method to explore configuration paths for the sustainable innovation of startups.

First, technical knowledge, market knowledge, a formal search, an informal search, organizational learning, and strategic flexibility are the main factors that affect the sustainable innovation in new ventures. However, they are not sufficient and necessary conditions to affect sustainable innovation—that is, the improvement in sustainable innovation of new ventures is not determined by one variable.

Second, the multiple interactions of resources and capabilities jointly affect the sustainable innovation in new ventures. Only relying on resources or enterprise capabilities cannot drive sustainable innovation. When startups have both resources and enterprise capabilities, sustainable innovation can be achieved.

Third, this study found that there are two pathways to drive the high sustainable innovation levels of startups in China. In the first path, a technical knowledge search, market knowledge search, organizational learning, and strategic flexibility are the core conditions, which is called the knowledge content-driven path. In the second path, a formal search, an informal search, organizational learning, and strategic flexibility are the core conditions, which is called the search method-driven path.

### 6.2. Theoretical Implications

This research has the following theoretical implications: First, it enriches the open innovation theory and expands the research boundary of the knowledge search. The previous research mainly focused on mature enterprises and reached the research conclusion of over-searching [31,102]. For startups, because they lack resources and rely more heavily on external knowledge, whether they have excessive search behaviors needs to be verified urgently [27]. In addition, in the existing research dimension construction, the duality of the knowledge search is relatively common—for example, search width and search depth [21,31], exploratory search and exploit search [103]. However, such a division approach does not solve the problem of what knowledge startups should search for, who should search for knowledge and how they should search for this knowledge. Therefore, this paper adopts a multi-angle division approach, combining the enterprise knowledge source (knowledge content) and the search method.

Second, it integrates the firm's "resources" and "capacity" frameworks and incorporates them into a research model. Most of the previous literature has addressed the impact of resource acquisition behavior or organizational capabilities on corporate competitiveness and performance or tried to open the black box. However, there is little research on the multivariate interaction of resources and capabilities. Therefore, based on the configuration perspective, this paper incorporates the knowledge search content, knowledge search methods, organizational learning capabilities, and strategic flexibility into a research framework.

Third, this paper presents the configuration paths that drive the sustainable innovation of startups. Most of the previous studies used regression analysis that obtained the marginal effects between variables and failed to draw a path to improve and reduce sustainable innovation. Therefore, this paper explores the integration of knowledge content, search methods, and organizational capabilities, uses the fsQCA method to analyze sample data, and obtains configuration paths that obtain and reduce the sustainable innovation of startups.

### 6.3. Management Enlightenment

The Fourteenth Five-Year Plan of China states that it is necessary to drive development with innovation and build the capacity for sustainable development in emerging industries. Acquiring the knowledge needed and improving their capabilities have enabled some

Chinese enterprises to develop from technological catch-up to technological leadership. Startups are more dependent on external resources than mature enterprises due to their lack of resources, poor R&D capabilities, and insufficient social relations. Therefore, external knowledge search activities have become essential for startups to acquire new knowledge and maintain competitive advantages. Meanwhile, due to the increased uncertainty in the external environment, startups need to improve their dynamic capabilities through learning and flexible thinking. Therefore, acquiring knowledge through knowledge search activities and improving organizational learning capabilities and strategic flexibility can improve sustainable innovation among startups. The management enlightenment of this paper is mainly reflected in the following aspects:

(1) New ventures should pay attention to external knowledge sources. It can be seen from this study that technical knowledge and market knowledge are the core to maintaining the sustainable innovation of startups. For them, there two main ways of acquiring knowledge, which are technological knowledge-dependent and market knowledge-dependent. In the technical knowledge-dependent type, technical knowledge is the core element, and enterprises need to cooperate with suppliers, universities, and research institutions to obtain the technical knowledge. In the market knowledge-dependent type, market knowledge is the core element, and enterprises need to obtain relevant market knowledge from organizations such as customers or competitors in order to change the existing business model or increase market share.

(2) New ventures should pay equal attention to formal and informal searches. It can be seen from the above research that the use of formal and informal search methods by enterprises is also an important factor in maintaining competitiveness. Startups can quickly acquire a large amount of new knowledge through formal search methods, and the introduction of new production equipment or production lines and the employment of relevant technical personnel have made the inflow of knowledge more obvious. However, using informal search methods is equally important, because only through informal search methods can we obtain tacit knowledge or make tacit knowledge explicit, which is more conducive to sustainable innovation.

(3) New ventures need to improve their organizational learning capabilities. To cope with the complex external environment, startups need to innovate continuously and develop into a learning organization. Looking at some leading enterprises of China, such as Ali, Tencent, and Huawei, we can see that their rapid development is inseparable from the learning and the formation of sustained competitive advantages. One step is to transform the acquired knowledge into the enterprise's innovation. The second step is to strengthen mutual benefit and trust among organizations, promote knowledge exchange and communication, and increase their knowledge transfer and sharing. The third step is to identify and discover new opportunities through the generation of new knowledge, expand the business scale, and enhance the sustainable innovation of the enterprise.

(4) New ventures need to improve their strategic flexibility. At present, the survival environment of startups is becoming increasingly turbulent. Measures to survive in this complex and changeable environment have become a primary factor in research. However, it is undoubtedly a good method to allocate resources flexibly and reorganize resources. First of all, startups should pay close attention to changes in the business environment and reduce market risks. Secondly, startups need to improve their management capabilities to increase the integration efficiency of internal and external resources. Finally, business managers should cultivate flexible strategic thinking and form organizational routines. In this way, startups can improve their strategic flexibility and strengthen their current operating conditions to maintain sustainable innovation.

### 6.4. Limitations and Future Research

Based on the open innovation and dynamic capabilities theory, this paper constructed an integrated framework of resource acquisition behavior and capabilities from the perspective of configuration, explored the configuration path that drives the sustainable innovation of startups, and obtained some theoretical and practical implications. However, this research still has the following limitations. First, there are limitations in the research area and sample selection. In the future, sample sources and sample sizes can be expanded, and more regional data can be collected as samples to improve the quality and applicability of the research. Second, this paper did not subdivide enterprises according to the industry to which they belong. Since the ability and knowledge search activities of enterprises are affected by industry factors, the manufacturing, information, and service industries can be subdivided to study the drivers of sustainable innovation in a certain industry. Third, there are other integration frameworks in terms of resource acquisition methods and corporate capabilities. In future research, other different variables can be selected for study.

**Author Contributions:** Y.L. designed the research and wrote this paper; H.Z. helped with data acquisition and analyzed the data. All authors have read and agreed to the published version of the manuscript.

**Funding:** This research received no external funding.

**Institutional Review Board Statement:** Not applicable.

**Informed Consent Statement:** Not applicable.

**Data Availability Statement:** Not applicable.

**Conflicts of Interest:** The authors declare no conflict of interest.

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
