# Peer review of "Driving Sustainable Innovation in New Ventures: A Study Based on the fsQCA Approach"

_sustainability, doi:10.3390/su14095738_

Round 1

Reviewer 1 Report

The detailed comments are as follows:

  1. The abstract lacks the time range of the research.
  2. Authors referring to literature and various research approaches indicate many new features of their study. However, it should be noted that the main purpose of the article is vague.
  3. The process of selecting the research sample requires a thorough explanation. How 450 companies were selected for research, how this applies to all companies in China.
  4. The article contains graphic elements (diagrams), but it is not known whether they are of their own authorship or come from the cited literature.
  5. The article definitely requires an editorial correction.

Author Response

Dear Reviewer:

Thank you for your comments concerning our manuscript entitled “Driving Sustainable Innovation in New Ventures:A study based on the fsQCA approach” (ID: 1704834). Those comments are all valuable and very helpful for revising and improving our paper, as well as the important guiding significance to our researches. We have studied comments carefully and have made correction which we hope meet with approval. Revised portion are marked in highlight in the paper. The corrections in the paper and the responds to your comments are as following:

Point 1:The abstract lacks the time range of the research.

Response 1: We have added the time range in the abstract.

Point 2:Authors referring to literature and various research approaches indicate many new features of their study. However, it should be noted that the main purpose of the article is vague.

Response 2: We have all rewritten the introduction according to your suggestion to understand this research's purpose clearly. First, in this section, we supplement the relevant literature to support the sufficiency of the argument; Second, we discuss the gaps in existing research and how we can narrow them. Finally, we describe the theoretical contributions of this paper and the questions we want to explore. This paper aims to explore the main factors affecting sustainable innovation in new ventures from the resource-capability integration view and the configuration paths that drive sustainable innovation.

Point 3:The process of selecting the research sample requires a thorough explanation. How 450 companies were selected for research, how this applies to all companies in China.

Response 3: First, we added specific details and explanations of the survey and implementation of the sample selection to ensure that it can be applied to all startups in China; Second, we describe the three top reasons for choosing this specific industry; Third, according to the characteristics of the questionnaire, we made a related table, see Table 1.

Point 4:The article contains graphic elements (diagrams), but it is not known whether they are of their own authorship or come from the cited literature.

Response 4: We've added sources to the  figures. Specifically, figure 1 is modified and proposed according to Sofka & Grimpe (2010)and Guo B & Guo JJ (2011); Figure 2 is the author's point of view based on the literature and related theories; Figure 3 is the research model based on theory and related literature.

Point 5:The article definitely requires an editorial correction.

Response 5: We have systematically revised this article. First, We have entirely rewritten the introduction to understand the purpose of this research clearly; Second, we have made a lot of revisions to the theory and model to provide theoretical support; Third, we add a robustness test at the end of the empirical analysis. Based on previous experience, we conducted robustness tests in two ways to ensure the reliability of the empirical results in this paper. Finally, we have made extensive revisions to our results and conclusions.

Reviewer 2 Report

The authors have conducted a serious study with the construction of an integrated structure of resources and capabilities in terms of configuration. Of particular note is the extended description of the theoretical and practical implications of the study results. In general, positively evaluating the article, it is worth highlighting a number of technical comments that require clarification or correction, for example:
1. Figures and tables do not have links to sources.
2. Table 4 is poorly formatted and it is not clear what the values ​​of the last two variables refer to.
3. Section 5.1 The conclusion looks very modest and can be expanded.

Author Response

Dear Reviewer:

Thank you for your comments concerning our manuscript entitled “Driving Sustainable Innovation in New Ventures:A study based on the fsQCA approach” (ID: 1704834). Those comments are all valuable and very helpful for revising and improving our paper, as well as the important guiding significance to our researches. We have studied comments carefully and have made correction which we hope meet with approval. Revised portion are marked in highlight in the paper. The corrections in the paper and the responds to your comments are as following:

Point 1: Figures and tables do not have links to sources.

Response 1: We've added sources to the  figures and tables. Specifically, figure 1 is modified and proposed according to Sofka & Grimpe (2010)and Guo B & Guo JJ (2011); Figure 2 is the author's point of view based on the literature and related theories; Figure 3 is the research model based on theory and related literature.

Point 2:Table 4 is poorly formatted and it is not clear what the values of the last two variables refer to.

Response 2: We have re-formatted Table 4 and explained the meaning of the last two variables in the note. “High-SI” represents the pathway to enhancing sustainable innovation;  “Low-SI” means the path to reducing sustainable innovation.

Point 3:Section 5.1 The conclusion looks very modest and can be expanded.

Response 3: We have made extensive revisions to our results and conclusions based on your suggestion. First, we rewrote the discussion of the results to make it more explicit; Second, we linked the study's findings to theory and literature in the talk of the results; Finally, around the purpose of this study, we have rewritten the Conclusions to ensure that the article is coherent.  

Once again, thank you very much for your comments and suggestions.

Reviewer 3 Report

please find the comments enclosed

Author Response

Dear Reviewer:

Thank you for your comments concerning our manuscript entitled “Driving Sustainable Innovation in New Ventures: A study based on the fsQCA approach” (ID: 1704834). And thank you so much for acknowledging this research and the work we do. In future research, we will continue to study the issue of resource-capacity integration. Your recognition is my most important asset on the road to scientific research.

Once again, thank you very much for your comments.

Reviewer 4 Report

The manuscript is well written, language is good, apart from a few minor grammatical and syntactic mistakes. The authors will surely find them when revising the manuscript. The manuscript also has a good elaboration of facts, good argumentation, and good explanation. -The work is an insightful academic endeavor, and in this case, it has resulted in some valuable conclusions. However, I have described some noteworthy points that the authors could address below.

The present manuscript has a week structure, and the authors are unable to support the conceptual framework with the support of the relevant theory. Most importantly, the methodology. A more accurate way to structure the theoretical section would be to separate the theoretical framework and the literature review from the hypothesis development. In this regard, the arguments for the Development of hypotheses should be grounded in theory engaged in the paper, also taking into account the peculiarities of the setting under investigation. In addition, while interesting, your research does not sufficiently articulate a contribution to the literature that is convincing: the theoretical contribution of your manuscript is narrowly defined.

Objective and positioning. The Introduction could do more to ground the paper's RQ in the debate and the related literature. In the actual version of the manuscript, scant attention is given to a theoretical derivation of the study's RQ and its actual positioning. Indeed, the lack of studies cannot be considered sufficient motivation for this study.
My suggestion is also to fully rewrite this section in order to answer the following questions: (i) Why is this topic relevant, and what is known about it? (ii) Which are the gaps you plan to address and how do you problematize them? (iii) How do you plan to close/address those gaps? (iv) Which are the main contributions of your study? . see the following

Sustainability in the Circular Economy: Insights and Dynamics of Designing Circular Business Models, Applied Sciences 12, no. 3: 1521. https://doi.org/10.3390/app12031521

Ambidextrous knowledge and learning capability: The magic potion for employee creativity and sustainable innovation performance. Sustainability 12, no. 10 (2020): 3966.
- In the Introduction, the 2nd paragraph must support a practical example, which supports the problem descriptions. A potential suggestion to the author/s is to support the importance of a title with some practical examples. However, it seems that too many things are bundled together, such that the logic and results are not very clear or convincing.

Theoretical framework and hypotheses. As already mentioned, I have several important reservations on the way the paper is conceptualized and the confounded way the topic is framed. Indeed, in my opinion, the theoretical framework is not well developed, and the theory the authors engage in the paper is not clearly explained. Please explain clearly the contributions to theory and practice because it is confusing. Moreover, one of the most important contributions in a literature review is to provide opportunities for future research and there is not here

Methodology. I must commend you for doing a good job with the statistical analysis. However, it is not clear how the setting and sample has been selected. I am particularly curious to know why the choice to consider such a specific industry. Also, I would recommend providing a table with the sample construction, showing how many observations you lose during the processes and how many unique firms you have in the final sample. More importantly, the variables' choice should be explained in light of the theory and the prior literature on the topic. At the moment, the authors do not provide any reference to prior studies measuring similar theoretical constructs. This should be mainly done to ensure the appropriateness and the robustness of both dependent and independent variables. Finally, you might want to provide a specific section to test the robustness of the analysis.

  1. Results and conclusion. The section devoted to the explanation of the results suffers from the same problems revealed so far. Your storyline in the results section (and conclusion) is hard to follow. Moreover, the conclusions reached are really far from what one can infer from the empirical results. The discussion should be rather organized around arguments avoiding simply describing details without providing much meaning. A real discussion should also link the findings of the study to theory and/or literature.

    5. Minor comments. There are some misprints and grammatical errors for what concerns the writing style. The manuscript would benefit from rewriting and professional proofreading.

    Hopefully, my suggestions will help you to improve this work, good luck!

Author Response

Dear Reviewer:

Thank you for your comments concerning our manuscript entitled “Driving Sustainable Innovation in New Ventures:A study based on the fsQCA approach” (ID: 1704834). Those comments are all valuable and very helpful for revising and improving our paper, as well as the important guiding significance to our researches. We have studied comments carefully and have made correction which we hope meet with approval. Revised portion are marked in highlight in the paper. The corrections in the paper and the responds to your comments are as following:

Point 1:Objective and positioning. The Introduction could do more to ground the paper's RQ in the debate and the related literature. In the actual version of the manuscript, scant attention is given to a theoretical derivation of the study's RQ and its actual positioning. My suggestion is also to fully rewrite this section according to my proposed modification. And the 2nd paragraph must support a practical example, which supports the problem descriptions. 

Response 1: We have all rewritten this part according to your suggestion and refer to ”Ambidextrous knowledge and learning capability: The magic potion for employee creativity and sustainable innovation performance.” First, in this section, we supplement the relevant literature to support the sufficiency of the argument; Second, based on your point of view, we address the reasons for studying this question in the first paragraph and add a practical example in the second paragraph. Then, we discuss the gaps in existing research and how we can narrow them. Finally, we describe the theoretical contributions of this paper and the questions we want to explore. This paper aims to explore the main factors affecting sustainable innovation in new ventures from the resources-capabilities integration and the configuration paths that drive sustainable innovation.

Point 2:Theoretical framework and hypotheses. In my opinion, the theoretical framework is not well developed, and the theory the authors engage in the paper is not clearly explained. Please explain clearly the contributions to theory and practice because it is confusing. Moreover, one of the most important contributions in a literature review is to provide opportunities for future research and there is not here.

Response 2: According to your comments, we have made a lot of revisions to the theory and model. First, we discuss the connotation of open innovation theory, knowledge-based view and dynamic capability theory and their theoretical support for this research; Second, a review on knowledge search, organizational learning, and strategic flexibility increases opportunities for future research; Third, in the model framework section, we add a large number of theoretical and practical discussions to support my model.

Point 3:Methodology. It is not clear how the setting and sample has been selected. I am particularly curious to know why the choice to consider such a specific industry. Also, I would recommend providing a table with the sample construction, showing how many observations you lose during the processes and how many unique firms you have in the final sample. More importantly, the variables' choice should be explained in light of the theory and the prior literature on the topic. At the moment, the authors do not provide any reference to prior studies measuring similar theoretical constructs. This should be mainly done to ensure the appropriateness and the robustness of both dependent and independent variables. Finally, you might want to provide a specific section to test the robustness of the analysis.

Response 3: First, we added specific details and explanations of the survey and implementation of the sample selection to ensure that it can be applied to all startups in China; Second, we described the three top reasons for choosing this specific industry; Third, according to the characteristics of the questionnaire, we made a related table, see Table 1; Fourth, we added robustness test at the end of the empirical analysis. Based on previous experience, we conducted robustness tests in two ways to ensure the reliability of the empirical results in this paper.

Point 4:The section devoted to the explanation of the results suffers from the same problems revealed so far. Your storyline in the results section (and conclusion) is hard to follow. Moreover, the conclusions reached are really far from what one can infer from the empirical results. The discussion should be rather organized around arguments avoiding simply describing details without providing much meaning. A real discussion should also link the findings of the study to theory and/or literature.

Response 4: Based on your suggestion, we have made extensive revisions to our results and conclusions. First, we rewrote the discussion of results to make it more explicit; Second, we linked the findings of the study to theory and/or literature in the results discussion; Finally, around the purpose of this study, we have rewritten the Conclusions to ensure that the article is coherent.

Point 5:Minor comments. There are some misprints and grammatical errors for what concerns the writing style. The manuscript would benefit from rewriting and professional proofreading.

Response 5: We have used the English editing services of MDPI, with professional corrections to grammar and writing style.

    Once again, thank you very much for your comments and suggestions.

Round 2

Reviewer 4 Report

- I appreciate the novelty of the author's contributions, but still, authors need to grasp deep ideas on developing the research gap in the Introduction.

-The last paragraph of the Introduction must be done a summary (resume) of the paper, i.e., a clear idea about what will be studied in the paper. I could not identify this. The last paragraph's purpose in the Introduction section is to summarize the main points, restate the paper's main idea, and show how the paper statements were proven. I think the paper will considerably improve and be a highly cited article.

-In the introduction, line 3-7, here I would recommend authors to explain, innovation in sustainability and its importance. see the following

Sustainability in the Circular Economy: Insights and Dynamics of Designing Circular Business Models, Applied Sciences 12, no. 3: 1521. https://doi.org/10.3390/app12031521

Ambidextrous knowledge and learning capability: The magic potion for employee creativity and sustainable innovation performance." Sustainability 12, no. 10 (2020): 3966.

-The structure of the Results and Discussion is not really logical, which results in several redundancies and in a text going several times back and forth between topics. Most interpretations are correct, but some must be questioned.

-Check the citations and references (one by one) if there is any missing information. Citations and references must be 100% accurate according to the journal guidelines.

-As a crucial improvement needed, without which the paper cannot be accepted, it is paramount to clearly show the relevance of your work to sustainability. Would you please consult the journal scope for more details?

-Moreover, please check all the references in the text carefully.

Author Response

Dear Reviewer:

    Thank you for your comments concerning our manuscript entitled “Driving Sustainable Innovation in New Ventures: A study based on the fsQCA approach” (ID: 1704834). Those comments are all valuable and very helpful for revising and improving our paper, as well as the important guiding significance to our research. We have studied the comments carefully and have made corrections which we hope meet with approval. Revised portions are marked in highlight in the paper. The corrections in the paper and the responses to your comments are as follows:

Point 1: The last paragraph of the Introduction must be done a summary (resume) of the paper, i.e., a clear idea about what will be studied in the paper.

Response 1: We have rewritten the last paragraph of the introduction and summarized the main points of this paper according to your comments.

Point 2: In the introduction, line 3-7, here I would recommend authors to explain, innovation in sustainability and its importance. 

Response 2: We have explained the innovation in sustainability and its importance.

Point 3: The structure of the Results and Discussion is not really logical, which results in several redundancies and in a text going several times back and forth between topics.

Response 3: We have revised the Discussion and removed redundant statements to make it more logical.

Point 4:  Please check all the references in the text carefully.

Response 4: We have checked all the references one by one and modified the format according to the journal guidelines.

       Moreover, we submitted the manuscript to “Special Issue: Innovation for Sustainable Business.” The need for sustainable development creates opportunities for businesses by implementing innovations designed to provide a competitive advantage to those who apply more sustainable practices and offer more sustainable products. To promote the effectiveness of innovations for sustainable business, there is an urgent need to understand the conditions that influence the success or failure of their implementation. This paper aims to explore the main factors affecting sustainable innovation in new ventures from the resources-capabilities integration and the configuration paths that drive sustainable innovation. We explore the conditions that affect the innovation for sustainable business at the micro-level. This is very consistent with the research in this special issue. In addition, we also employ an emerging management method for the implementation of sustainability-oriented innovation.

     Once again, thank you very much for your comments and suggestions.